# Impact assessment for just transition of protein production systems

Harpinder Sandhu[1]*, Amie Sexton[1], Priyambada Joshi[1], Lee Recht[2],
Ayon Chakraborty[3], Sukhbir Sandhu[4]

1 Ararat Jobs and Technology Precinct, Institute of Innovation, Science and Sustainability, Federation University Australia, Mount Helen, Victoria, Australia, 2 Israel Innovation Authority, Jerusalem, Israel, 3 Institute of Innovation, Science and Sustainability, Federation University Australia, Mount Helen, Victoria, Australia, 4 Centre for Workplace Excellence (CWeX), UniSA Business, University of South Australia, Adelaide, South Australia, Australia

* Harpinder.sandhu@federation.edu.au

## Abstract

Understanding the impacts of protein production systems is necessary to plan the just transition of food systems. We analysed 285 studies to assess the impacts of 13 protein systems across 25 indicators under five key categories—natural capital, human capital, social capital, produced capital and governance. Nine protein systems (regenerative, organic, rangelands, free-range poultry, sustainable energy cultivated meat, conventional energy cultivated meat, mixed grains and livestock, pastoralists and plant-based) have overall positive impacts across all five categories. In comparison, four protein systems have negative impacts (small-scale beef, caged poultry, industrial pork, and confined feeding operations). We then used this in-depth assessment to develop five 'what if' future scenarios to track and assess the transition of protein production systems to 2050. Rapid reduction of industrial production may contribute to a just and inclusive transition of protein production systems. This assessment can help reduce risks associated with negative impacts and assist in governing and managing protein production systems towards long-term sustainability.

## Introduction

Meeting the increasing demand for diverse and nutritious foods by a projected human population of 10 billion by 2050 and the consequent social, economic and environmental impacts constitute a significant challenge for global food systems [1–5]. Not surprisingly, 12 of the United Nation's 17 Sustainable Development Goals (SDGs) are linked to food and agriculture [6,7]. To achieve these SDGs, the global food and agricultural systems must be transformed by evaluating and tracking their progress across social, health, governance, environmental and economic indicators [8–10]. While a large number of studies evaluate specific indicators in specific production systems, there is limited scientific literature that enables comparing multiple protein production systems across multiple social, environmental and economic indicators.

**Data availability statement:** Excel database is uploaded as Supporting information.

**Funding:** HS was awarded the grant by Federation University Australia and Aleph Farms Joint Research grant G2355. https://www.federation.edu.au/https://aleph-farms.com/ Priyambada Joshi, PhD candidate received stipend/financial support from this joint research grant by Federation University Australia and Aleph Farms. The funders had no role in study design, data collection and analysis, decision to publish, or preparation of the manuscript.

**Competing interests:** Co-author Dr Lee Recht was previously employed at Aleph Farms during the data analysis and writing of the article. Dr Recht no longer works at this organisation.

In this study, we assess 13 protein production systems by comprehensively evaluating their impacts across natural, human, social, produced capital and governance indicators. Protein production systems are defined as those that aim to produce protein; in this study, we focus on beef, pork, poultry, and two emerging alternatives (plant-based and cultivated meat). The in-depth impact assessment is then used to develop future scenarios to enable a just transition of the protein production systems. Understanding the social, environmental and economic impacts of various protein production systems is necessary to plan a just transition towards sustainable, inclusive and equitable systems for livestock farmers, the meat sector, investors, and consumers [3,8]. A comprehensive impact assessment can help practitioners and policymakers adopt sustainable practices and policies that can allow them to develop appropriate responses for continuous improvements in each system over time. These improved practices and policies can contribute to a just transition of the sector [11–13]. The just transition targets zero hunger, environmental sustainability, climate action, building climate resilience, job creation, poverty reduction and social equity in global protein production systems [14].

Global agriculture and food production systems meet the current calorific requirements for over 8 billion people worldwide [15,16]. However, aggregated food demand is set to increase by 70 per cent by 2050 [15]. At the same time, the demand for protein is growing rapidly, especially in developing and emerging economies. Since the domestication of animals for food sources around 10,000 years ago, animal protein has played an increasingly significant role in the human diet [17,18]. Animal products, including meat, eggs and dairy, are well known as a primary source of complete protein and B12 and are consumed widely. Plant-based foods such as fruits, vegetables, grains, lentils, nuts, and seeds are also consumed in many parts of the world as a primary source of protein. Currently, livestock products provide only 34% of the global protein demand, but the demand for meat, eggs, and dairy will likely grow by nearly 80 per cent by 2050 [15,19]. There has been an increase in the consumption of processed plant-based protein with attributes similar to those of conventional meat in taste and flavour [20].

The global meat sector, including subsistence and industrial scale production worldwide, generates $1.3 trillion annually—40 per cent of the global value—of agricultural output and supports the livelihood and food security of about 1.7 billion people [21–23]. However, the global meat sector occupies 75 per cent of all agricultural land, uses more than 40 per cent of agricultural water, contributes to climate change with 20–30% greenhouse gas emissions, pollutes waterways, causes biodiversity loss, and is facing increasing scrutiny due to animal welfare concerns [24–26].

An awareness of these negative impacts is increasingly driving sustainable initiatives at farm and industry levels. Rising pressure from consumer groups, and regulation and environmental protection agencies is leading the producers, the livestock industry and the companies to be innovative [27–30]. One such innovation led by food companies is cellular agriculture, which harnesses cells to grow animal products in controlled environments. Cultivated meat is an application of animal cells, which is a new animal product category. Like conventional meat, cultivated meat contains

animal muscle, fat, and collagen. However, these elements are grown directly from animal cells rather than by raising and slaughtering an animal. Our study includes this innovative protein production system among the 13 protein systems that we investigate.

The majority of existing studies that compare different protein production systems mainly focus on only the environmental dimensions [31–33]. There is a lack of research that focuses on a comprehensive assessment of relevant social, environmental and economic dimensions across multiple protein production systems. This study addresses this gap and highlights the impacts of various protein production systems on a comprehensive set of indicators that are key for the sustainability of the protein sector. In doing so, it enables sustainability, inclusivity, and equity within protein production systems.

In this study, we selected 13 different protein production systems, including seven beef systems (pastoralist, small-scale, mixed grains and livestock, organic, confined feeding operations, rangelands and regenerative), one industrial pork system, two poultry systems (caged and free-range), one plant-based alternative and two types of cultivated meat (conventional energy and renewable energy). We assessed their impacts on 25 multi-dimensional indicators across five key categories – natural, human, social, and produced capital and governance [1,2,16]. We then employed this assessment to develop 'what if' future scenarios using the current and future speculative market share of four leading systems to track the transition of protein production systems to 2050.

This study emphasises just transition, so instead of classifying protein production systems into a simplistic and binary sustainable versus unsustainable dichotomy, we position them along a continuum from unsustainable to very sustainable and show ways in which negative impacts can be reduced and improvements made over time [34]. This study makes three main contributions. First, it provides an up-to-date assessment of multiple protein production systems, including newer and innovative plant-based protein and cultivated meat, across multiple indicators. This adds to the growing knowledge of the social and health impacts of protein production systems [1,35]. Second, in addition to providing comprehensive impact assessment data across 13 protein production systems, we then use this data to develop scenarios for the just transition of the meat sector so that appropriate responses can be developed in practice, policy and research for the sustainable transformation of agrifood systems [36]. Third, our research also provides methodological advancements through integrating a systematic literature review and the Delphi approach. Related methodological integrations have been used in medical and social sciences to address complex research questions that need large amounts of quantitative and qualitative data collected across diverse sources, the interpretation of which needs expert assessment to ensure reliability and validity [37,38].

## Methods

### Protein production systems

We selected 13 production systems across five different types of proteins that represent the protein sector globally. These are beef, pork, poultry, plant-based and cultivated meat systems. Beef, pork and poultry production accounts for 21, 30 and 38 per cent, respectively, of the total meat produced globally [35]. Their impacts have been widely studied globally. The production and consumption of plant-based proteins are growing worldwide. Similarly, cultivated meat is being developed using biotechnology and engineering to address some of the challenges of conventional meat production. Therefore, we included plant-based protein and cultivated meat (conventional and renewable energy) as three innovations in the protein sector alongside variants of beef systems (one traditional, four industrial and two extensive), industrial-scale pork production and two types of poultry systems (caged and free-range). We focused on a) the country level for traditional, small-scale, mixed systems, confined feeding operations, and rangelands, b) the regional level (Europe) for organic beef systems and free-range poultry, and c) the global level for regenerative, plant-based and cultivated meat systems. The country-level systems are included because they are well established and there are many publications that capture their

impacts relevant to a specific country. At the regional level (Europe), organic beef production and free-range poultry follow similar practices and policies. In contrast, regenerative, plant-based and cultivated meat systems are recent innovations and not enough primary studies have been conducted to describe or classify them to a given country or regional level. Therefore, these systems are included as a representation of the global level. Given the amount and quality of data and information available on these systems, they are comparable for various impacts. S1 Table (Supporting Information) provides a classification and description of these 13 systems.

## Impact categories

We identified indicators across five key categories (natural capital, human capital, social capital, produced capital and governance) associated with the protein production systems by following a comprehensive framework (the 'TEEBAgri-Food' framework) developed by the United Nations Environment Program to capture and measure all social, economic and environmental impacts—both negative and positive—in global agriculture and food systems [1,2]. This framework is increasingly being used to identify and value natural capital (e.g., well-functioning biodiversity and ecosystems), human capital (e.g., skills and knowledge), social capital (e.g., societal interactions, relationships, formal and informal institutions) and produced capital (e.g., finance and machinery) associated with agriculture and food systems. We also included another category—governance based on the current literature focused on environmental, social, and governance (ESG) reporting [39]. The agriculture and food sectors increasingly use ESG reporting to demonstrate their social and responsible agenda [40,41]. Examining their activities' impacts on the social, natural, and economic environment is necessary to manage just transition. In this study, we selected 25 multi-dimensional indicators across five broad categories (S2 Table, Supporting Information) based on the comprehensive literature on ESG and applications of the TEEBAgriFood framework in agriculture and food systems. We focused on the protein production systems, not the product level or value chain assessment.

## Data analysis

In this study, we used the data triangulation method to assess the impact of protein production systems. Using multiple sources of information enhances the study's reliability and validity via triangulation [42], with the range of perspectives creating a holistic picture. The study included a systematic review of the literature, data interpretation by the authors on a scale, and expert assessment involving two iterations using the Delphi approach [43].

**Systematic literature review.** We conducted a systematic literature review of published studies up to January 2025 using the Preferred Reporting Items for Systematic Reviews and Meta-Analyses (PRISMA) method [44]. To ensure comprehensiveness, we used 25 impact categories and 13 protein production systems as search strings to search the Web of Science database of literature published in English (see S3 Data, Supporting Information for the list of search strings). We did not specify a start date so that we could be as comprehensive as possible in our literature search. A total of 9,656 peer-reviewed articles were found with these keywords. These articles were further screened for 1) at least one clearly defined impact indicator and 2) quantifiable information about one production system. This narrowed our selection to 282 articles. However, some indicators were not captured in this assessment. We then conducted a Google Scholar search using the same search strings used for the Web of Science search and used the above screening method to select 72 relevant studies. Out of total 354 studies, we removed 69 duplications; the final number of studies used in this assessment was 285. Several studies are included for each system, and each indicator with full reference and key observations is provided in S3 Data (Supporting Information).

**Impact scale.** The 285 studies were further read and analysed to extract information about 25 selected indicators for each of the 13 production systems. No single study provided a comprehensive comparison of all systems for each indicator. To collate such vast data on 25 multi-dimensional indicators and 13 systems across different countries into a single unit, we developed a 10-point scale of −5 to +5 indicating extremely negative to extremely high positive impacts

([S4 Table](), Supporting Information). This scale allowed us to rigorously capture the multi-dimensional impacts of different production systems. A set of papers relevant to each indicator and each production system were read carefully. We also considered multiple studies out of the selected 285 studies that provided quantitative data for some indicators. We then extracted quantifiable information on each indicator for different protein production systems (see [S3 Data](), Supporting Information, for details). Then, we assigned a comparative unit ranging from −5 to +5 to compare each system for the selected indicator.

As an illustrative example, to enumerate the unit value of the impact of protein production systems on natural capital (indicator 1: net GHG emissions), we first read relevant studies identified during the systematic literature review for this indicator for each of the 13 systems (see Excel Database). Then, the system with the highest amount of GHG emissions was assigned a −5 unit value (confined feeding operations, US) as this is a negative impact. Others were given relative unit values based on their net GHG emissions (−3 for small-scale, China and industrial pork, China). For human capital (indicator 10: skills and knowledge of workers, training), we read selected studies for this indicator for each of the 13 systems (see Excel Database). Then, the system with the highest level of skills and training required was assigned a +5 unit value (organic Europe and regenerative global) as this positively impacts production. Others were given relative value based on assessments (+4 for rangelands in Australia and free-range poultry in Europe).

A table showing all impacts across 25 indicators in 13 protein production systems was generated from this assessment, with cells showing a unit value between −5 and +5. The main reason for allocating NA for various indicators under two cultivated meat systems is based on the current and up-to-date literature and consultation with the experts as detailed in the methodology section. Data analysis (data triangulation method) for impact assessment was used including systematic literature review of 285 studies, data interpretation by authors and then consultation with global experts in two iterations using Delphi approach. The indicators that show NA under cultivated meat systems are not applicable by definition. For example, indicator 3 (waste generation on farm), indicator 6 (loss of agri-enviro biodiversity on farm), indicator 9 (soil contamination), indicator 12 (health of farmers), indicator 13 (heath of farm and production workers) etc., are not relevant for cultivated meat production because the meat is produced in bioreactors in a laboratory or industrial setup. Indicators such as those related to production (indicators 20–23) are NA due to the lack of data, as cultivated meat is currently not being produced commercially at a scale to assess such impacts. The assessment is based on up-to-date literature up to January 2025.

**Delphi approach.** To generate consensus amongst global experts on this assessment and ensure the reliability and validity of these results, we used the Delphi approach and engaged with experts in two iterations. The objective of this process was 1) to verify the results and our interpretation from the systematic literature review to a scale, and 2) to modify these results after each expert provided their comparative assessment based on their research experience and understanding of each system. We selected 137 experts from university academics, scientists, consultants, and industry experts representing all 13 protein production systems worldwide. These experts were selected based on their publication record, reputation in their research field (h-index), job title, and relevant institution as per their publicly available profiles. Human ethics approval was obtained on 19 July 2022 from the Federation University Human Ethics Approval Committee before contacting experts between 1 August 2022 and 30 June 2023. An email explaining the research project and the impact assessment based on a systematic review and the impact table in Excel format was sent to each expert. A completed response was considered as consent to participate, as explained in the email addressed to the experts. We received 25 responses, representing an 18 per cent response rate. We received seven responses (about 5 per cent) that returned the completed assessment with modifications and justification for each indicator across different production systems. These seven experts were from industry, research institutes, the United Nations, and the university sector and have considerable collective expertise in commenting on 13 diverse protein production systems. The other 18 experts provided detailed comments but did not complete the assessment table. We then modified our assessment based on these responses and finalised the impact table by including all returned responses. We again shared this improved impact

table with the seven experts and provided another opportunity to modify it. All experts agreed with this revised impact assessment. We then finalised the multi-dimensional impacts across 13 protein production systems. The strict protocol we followed ensures replicability of our results in future research carried out under similar conditions. The primary round of literature review (n = 175 studies) was completed in December 2022. Ten additional studies were identified in 2023 (n = 3) and 2024 (n = 7) and included in the literature, resulting in a total of 285 studies. These studies did not change the findings, so they were not subjected to expert opinion.

### Scenarios for protein production futures

After finalising the multi-dimensional impact assessment of 13 protein production systems based on data up to 2025, we developed 'what if' future scenarios for the transition of protein production up to 2050. We included currently dominant industrial-scale production responsible for 68.6 per cent of the total protein production and focused on sustainable systems as a leading system for transition. The group termed 'others' in the scenarios includes regenerative, organic, mixed livestock, rangeland systems, and small-scale production, accounting for about 30 per cent of the total protein, with plant-based alternatives at 1.4 per cent. We selected four protein production systems—industrial, others, plant-based protein and cultivated meat—based on their current and future speculative market share, overall impacts and natural capital impacts based on our assessment. We developed five 'what if' scenarios by assuming no, slow, moderate, fast, and aggressive changes in how protein can be produced in the future. We kept the 'no change' scenario as a base for business as usual with no change in the impacts or market share (based on production) in future. The scenarios were developed based on this, and then their impacts were estimated using the current impact assessment as a baseline of 2025. Based on the potential changes in the market share of four different systems, we estimated their overall and natural capital impacts to 2030, 2040 and 2050. These scenarios included modelling the protein production based on various assumptions from 2025 to 2050. These assumptions were (1) S1 (slow transition: industrial scale production decreases by 1 per cent per year, others increase by 0.25 per cent per year, plant-based increases by 0.5 per cent per year, cultivated increases by 0.25 per cent per year), (2) S2 (moderate transition: industrial scale production decreases by 3 per cent per year, others increase by 1 per cent per year, plant-based increases to 1 per cent per year, cultivated increases by 1 per cent per year), (3) S3 (aggressive transition: industrial scale production decreases rapidly, others increase to 40 per cent by 2030 and then decrease slightly, plant-based increases rapidly, cultivated increases by 1 per cent per year by 2030 then the rate increases), (4) S4 (aggressive transition towards industrial systems: industrial scale production increases by 0.75 per cent per year, others decrease by 1.25 per cent per year, plant-based increases by 0.25 per cent per year, cultivated increases by 0.25 per cent per year). This initial modelling estimated the market share through 2030, 2040 and 2050. The impact assessment scores from the baseline of 2025 were then used to calculate the impacts for this period and plotted on a graph to show the transition of protein production systems.

## Results

### Analysis of literature

The literature analysis yielded 285 studies used in this study for multi-dimensional impact assessment. Fig 1 provides an overview of the systematic literature review. We plotted these studies against the year of publication. We found an increasing trend in the number of studies published from 2001 onwards (Fig 2a), peaking in 2021. We analysed this literature based on five key categories: natural, human, social, produced capital and governance. We found a higher number of studies focused on the impacts of protein production systems on natural capital (n = 134). This was followed by produced capital (n = 57), human capital (n = 48), social capital (n = 43) and governance (n = 43) (Fig 2b). Amongst each of the 25 indicators, the highest number of studies were on net GHG emissions, followed by the influence of policies and regulations, land use change, consumers' health, and animal health (Fig 2c). A disproportionate number of studies focused on

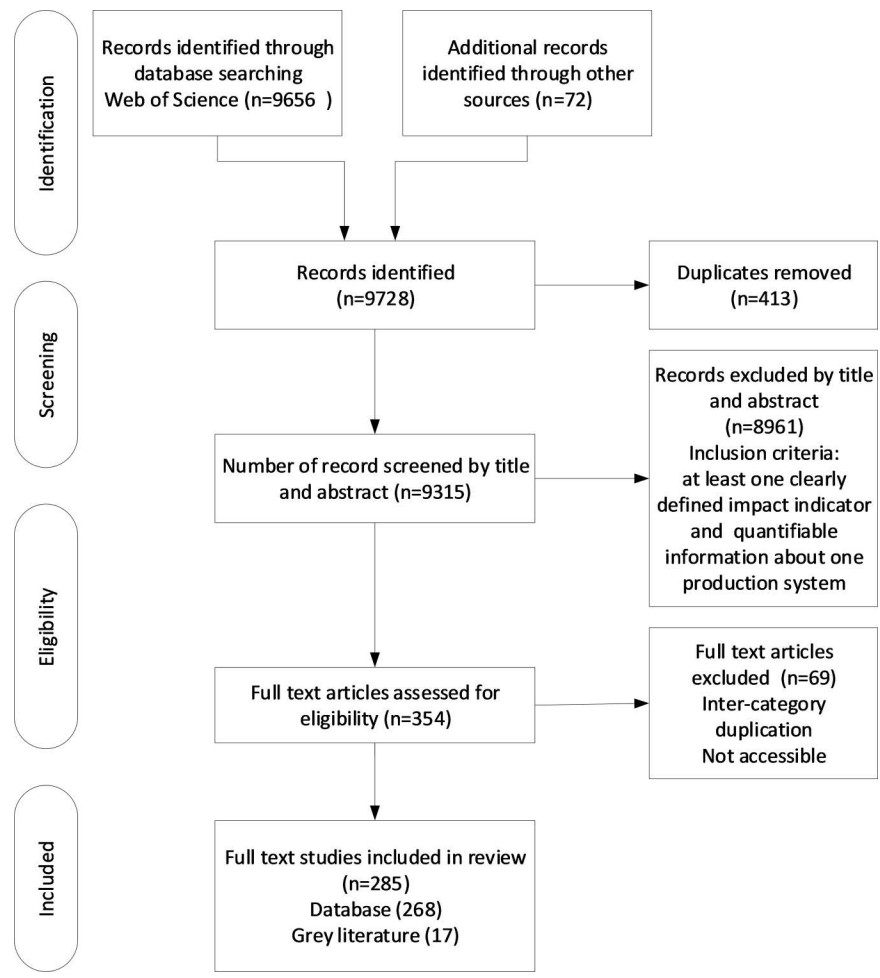

**Fig 1. A flow chart showing an overview of the systematic literature review.**

beef production systems, with pastoralists as the leading system, followed by rangelands, small-scale beef systems in China and confined feeding operations in the US, followed by pork and poultry systems (Fig 2d).

## Impact assessment

The multi-dimensional impacts of 13 different protein production systems are provided in Table 1. Under the natural capital category (nine indicators), confined feeding operations scored the lowest (−43 out of a possible score between −45 and +45). This production system has an extremely negative impact (−5) on five indicators in particular: waste generation on farms, air pollution, loss of livestock biodiversity, land use change, land degradation and soil contamination. This was followed by the industrial pork system (−33) and small-scale beef (−26). None of the 13 protein production systems had an extremely positive impact on all nine indicators in the natural capital category.

In the human capital category, the industrial pork and caged poultry systems produced overall negative impacts (−14 out of a possible score between −25 and +25), followed by confined feeding operations (−12). No system had an extreme negative impact on any of the five indicators. However, there was very high negative impact (−4) recorded for workplace health and safety standards, health of farmers, health of farm workers, and health of consumers. Organic and

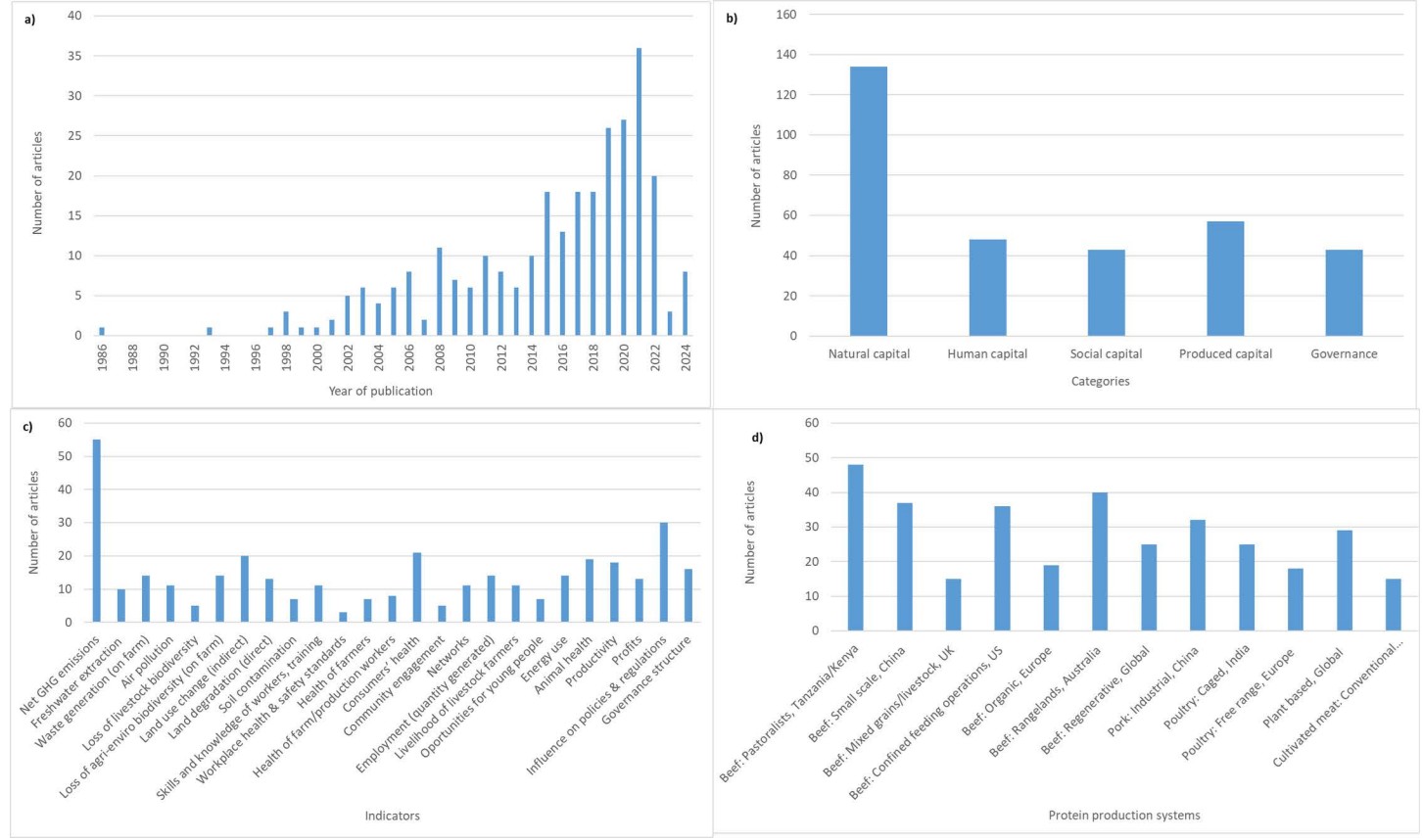

**Fig 2. a) Number of articles by the year of publication, (b) five main categories, (c) each indicator, and (d) the type of protein production system.**

regenerative systems had an extreme positive impact (+5) on skills and knowledge of workers, while regenerative and conventional energy cultivated meat had an extreme positive impact on workplace health and safety standards.

For the social capital category (five indicators), regenerative scored the highest (+17 out of a possible score between −25 and +25). Notably, none of the systems scored an extreme positive impact for any indicator in this category. Community engagement and opportunities for young people were scored at (+4) in regenerative systems. A high negative impact (−3) was recorded for community engagement in industrial pork and caged poultry systems.

Under the produced capital category (four indicators), regenerative led (+14 out of a possible score between −20 and +20), followed by rangelands, organic, and free-range poultry systems. The confined feeding operations system had a high negative impact on energy use (−4). The regenerative system scored an extreme positive impact (+5) for productivity, followed by rangelands, organic, mixed and confined operations systems (+4).

For the governance category (two indicators), cultivated meat conventional and renewable energy scored the highest (+8 out of a possible score between −10 and +10), followed by rangelands, plant-based and organic systems.

Fig 3 shows the rank of each system based on its overall score. Nine systems (regenerative, organic, rangelands, free-range poultry, sustainable energy cultivated meat, conventional energy cultivated meat, mixed grains and livestock, pastoralists, and plant-based protein) have overall positive impacts. Regenerative scored the highest points (+63 out of the impact range of −125 to +125), followed by organic (+45) and rangelands systems (+36). In comparison, four systems

**Table 1. Impact assessment of different protein production systems 2025 (0: No information, NA: Not applicable in the impact categories).**

| Indicators | Traditional beef | Industrial scale beef | | | | Extensive beef | | Pork | Poultry | | Plant-based | Cultivated meat | | Average of all systems |
|---|---|---|---|---|---|---|---|---|---|---|---|---|---|---|
| | Pastoralists, Tanzania/Kenya | Small-scale, China | Mixed grains and livestock, UK | Organic, Europe | Confined feeding operations, US | Range-lands, Australia | Regenera-tive, Global | Industrial pork, China | Caged poultry, India | Free-range poultry, Europe | Plant-based, Global | Conventional energy, Cultivated meat, Global | Sustainable energy, Cultivated meat, Global | |
| **Natural capital** | | | | | | | | | | | | | | |
| 1 Net GHG emissions | -2 | -3 | -2 | -1 | -5 | -2 | -1 | -3 | -2 | -1 | -2 | -3 | -1 | |
| 2 Freshwater extraction (water pollution) | -1 | -3 | -2 | -2 | -4 | -1 | 0 | -3 | -2 | -1 | -3 | 1 | 1 | |
| 3 Waste generation (on farm) | -1 | -3 | -2 | 0 | -5 | -2 | 0 | -4 | -3 | -1 | -1 | NA | NA | |
| 4 Air pollution | -2 | -3 | -2 | -2 | -5 | -1 | 0 | -4 | -2 | -1 | -1 | 1 | 2 | |
| 5 Loss of livestock biodiversity | 3 | -3 | -2 | 0 | -5 | -1 | 1 | -4 | -4 | -1 | NA | -1 | 0 | |
| 6 Loss of agri-enviro biodiversity (on farm) | 0 | -2 | -1 | 3 | -4 | 1 | 3 | -3 | -3 | 0 | -3 | NA | NA | |
| 7 Land use change (indirect) | 2 | -3 | -1 | 0 | -5 | -1 | 2 | -4 | -3 | 0 | -1 | 1 | 1 | |
| 8 Land degradation (direct) | -2 | -3 | -2 | 3 | -5 | 0 | 3 | -4 | -3 | 0 | -2 | 3 | 3 | |
| 9 Soil contamination | 0 | -3 | -1 | 1 | -5 | 1 | 2 | -4 | -3 | 0 | -3 | NA | NA | |
| **TOTAL (of a possible max score +/-45)** | -3 | -26 | -15 | 2 | -43 | -6 | 10 | -33 | -25 | -5 | -21 | 2 | 6 | -12 |
| **Human capital (individual)** | | | | | | | | | | | | | | |
| 10 Skills and knowledge of workers, training | 2 | 1 | 3 | 5 | 2 | 4 | 5 | 1 | 1 | 4 | 2 | 2 | 2 | |
| 11 Workplace health and safety standards | 1 | -2 | 3 | 4 | -3 | 4 | 5 | -4 | -4 | 4 | 3 | 5 | 4 | |
| 12 Health of farmers | -2 | -2 | -1 | 1 | -3 | 1 | 2 | -4 | -4 | 1 | -1 | NA | NA | |
| 13 Health of farm/production workers | -2 | -2 | -1 | 1 | -4 | 1 | 2 | -4 | -3 | 0 | -1 | NA | NA | |
| 14 Consumers health | 2 | -3 | -1 | 2 | -4 | 2 | 2 | -3 | -4 | 2 | 0 | NA | NA | |
| **TOTAL (of a possible max score +/-25)** | 1 | -8 | 3 | 13 | -12 | 12 | 16 | -14 | -14 | 11 | 3 | 7 | 6 | 2 |

*(Continued)*

**Table 1.** (Continued)

| # | Indicator | Traditional beef | Industrial scale beef | | Extensive beef | | Pork | Poultry | | Plant-based | Cultivated meat | | Average of all systems |
|---|-----------|:---:|:---:|:---:|:---:|:---:|:---:|:---:|:---:|:---:|:---:|:---:|:---:|
| **Social capital (collective)** | | | | | | | | | | | | | |
| 15 | Community engagement | 3 | −1 | 3 | 3 | 4 | −3 | −3 | 3 | 1 | −1 | −1 | |
| 16 | Networks | 1 | 2 | 3 | 2 | 3 | 1 | 1 | 3 | 2 | 2 | 2 | |
| 17 | Employment (quantity generated) | 2 | 2 | 3 | 3 | 3 | 1 | 1 | 2 | 2 | 0 | 0 | |
| 18 | Livelihood of livestock farmers | 2 | 2 | 2 | 3 | 3 | 0 | 1 | 2 | NA | NA | NA | |
| 19 | Opportunities for young people | 0 | 0 | 1 | 2 | 4 | 0 | 1 | 2 | 2 | 3 | 3 | |
| | **TOTAL (of a possible max score +/-25)** | 8 | 5 | 12 | 13 | 17 | −1 | 1 | 12 | 7 | 4 | 4 | 7 |
| **Produced capital** | | | | | | | | | | | | | |
| 20 | Energy use | 1 | −2 | −2 | −2 | 0 | −3 | −1 | 0 | −2 | NA | NA | |
| 21 | Animal health | 0 | −2 | 2 | 3 | 3 | −3 | −3 | 3 | NA | NA | NA | |
| 22 | Productivity | 0 | 3 | 4 | 4 | 4 | 3 | 2 | 3 | 4 | NA | NA | |
| 23 | Profits | 0 | 2 | 4 | 4 | 3 | 3 | 2 | 3 | 4 | NA | NA | |
| | **TOTAL (of a possible max score +/-20)** | 1 | 1 | 8 | 9 | 10 | 0 | 0 | 9 | 6 | 0 | 0 | 5 |
| **Governance** | | | | | | | | | | | | | |
| 24 | Influence on policies and regulations | −1 | 2 | 3 | 4 | 3 | 2 | 2 | 3 | 4 | 3 | 3 | |
| 25 | Governance structure | 2 | 1 | 4 | 2 | 3 | 1 | 1 | 3 | 3 | 5 | 5 | |
| | **TOTAL (of a possible max score +/-10)** | 1 | 3 | 7 | 6 | 6 | 3 | 3 | 6 | 7 | 8 | 8 | 5 |
| | **OVERALL TOTAL (of a possible max score +/-125)** | 8 | −46 | −25 | 36 | 45 | −45 | −35 | 33 | 63 | 21 | 24 | 7 |

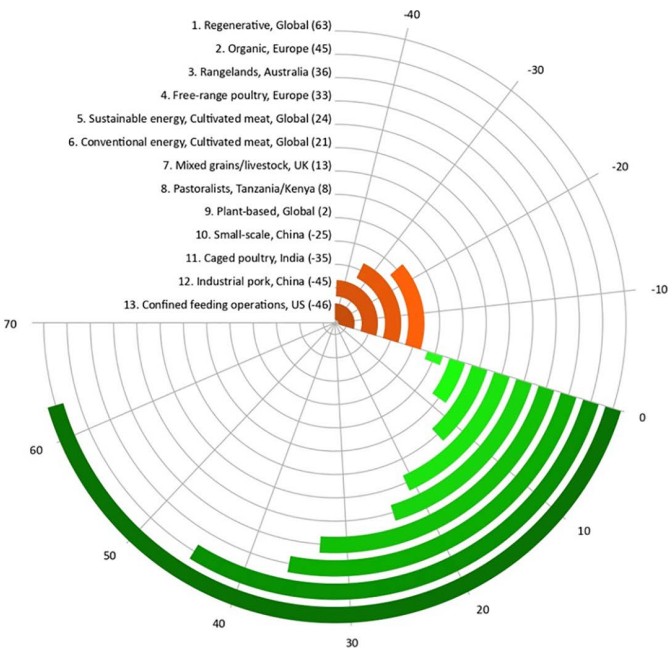

**Fig 3. Ranking of all protein production systems based on their overall impacts, including natural, social, human, produced capital and governance. Value: -125 to +125.**

have negative impacts (small-scale beef, caged poultry, industrial pork, and confined feeding operations). Out of these, confined feeding operations scored (−46), industrial pork (−45), poultry (−35) and small-scale beef and plant-based systems (−25).

Further examination of the impacts in each of the five broad categories shows that for natural capital impacts, only four systems have positive impacts (regenerative, sustainable energy cultivated meat, conventional energy cultivated meat, and organic beef). The rest of the nine systems have negative impacts (Table 1). For human capital, four systems have negative impacts (small-scale beef, confined feeding beef operations, industrial pork, and caged poultry, Table 1). All systems positively impact social capital, except industrial-scale pork production (Table 1). In produced capital, all systems have positive impacts; however, cultivated meat is not yet in full-scale commercial production (Table 1), and therefore, we were unable to assess its impact in this category. For governance, all systems have positive impacts (Table 1).

### Protein production scenarios

We developed five future scenarios to illustrate just transition of the protein production systems. These scenarios are based on the current and projected market share for industrial, regenerative, plant-based protein and cultivated meat, as shown in Fig 4. The current (no change) scenario does not result in any change in the future. It assumes no change in the current market share of the four production systems. Slow change transition in scenario 1 results in industrial-scale production dropping to 50 per cent by 2040 and 40 per cent by 2050 from the current 68.6 per cent market share. Other production systems grow to 37 per cent, plant-based protein grows to 15.4 per cent, and cultivated meat captures 7 per cent of the market share by 2050. In the moderate change scenario 2, industrial-scale production drops to 14.6 per cent by 2040 and then to 1 per cent by 2050, others plus regenerative systems grows to 41 per cent, plant-based protein reaches 30 per cent, and cultivated meat takes up 28 per cent of the total market share by 2050. Fast change scenario 3 results in a rapid reduction in industrial scale production to 1 per cent by 2040, while the other three systems will take up

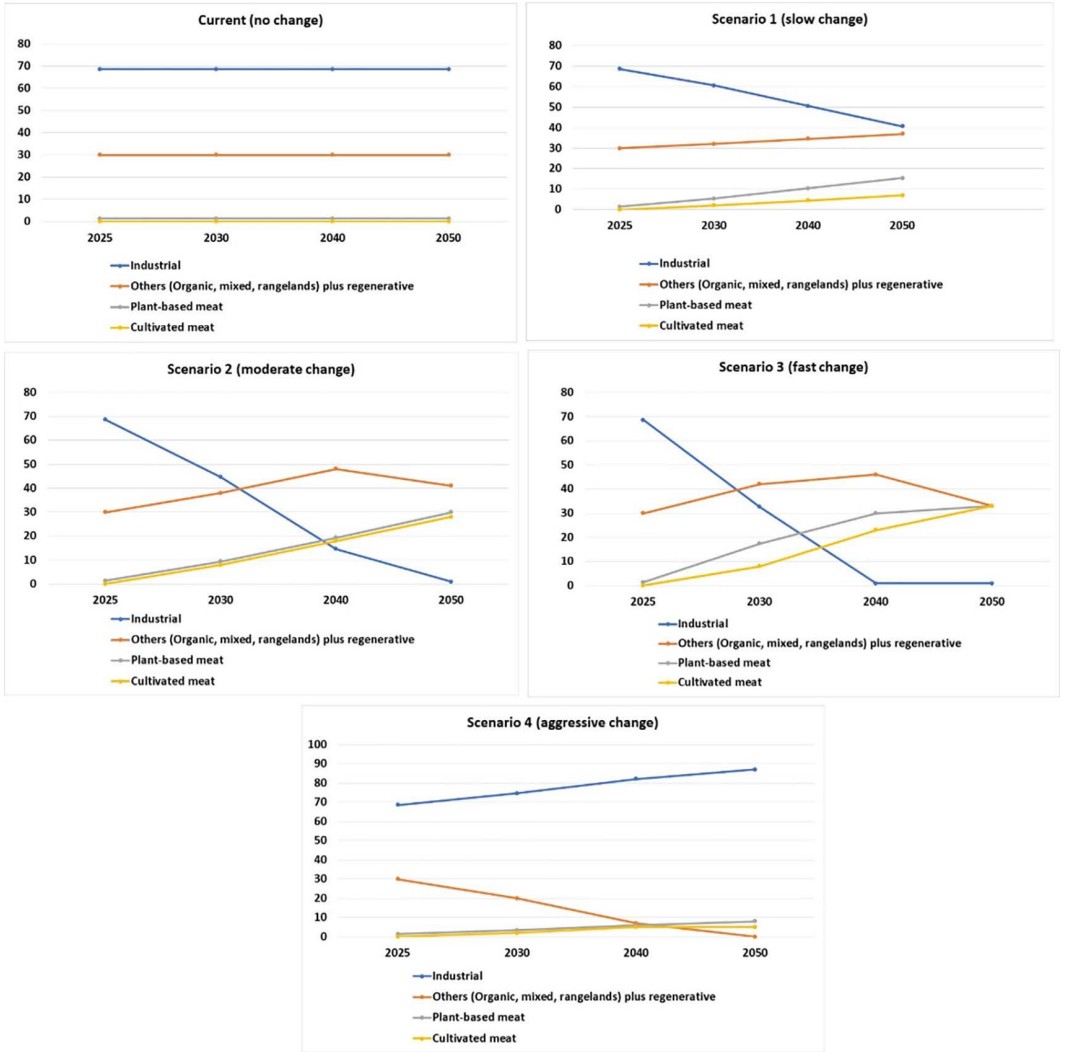

**Fig 4. Impacts (value: −125 to+125) of different protein production systems in five scenarios based on future market share projections of four leading protein production systems from 2025 to 2050.**

an equal share of 33 per cent each by 2050. In aggressive change scenario 4, the industrial production systems grow rapidly. It reaches up to 87 per cent by 2050, other plus regenerative systems decline to 7 per cent by 2040 and then decline to negligible by 2050, while plant-based protein reaches up to 8 per cent and cultivated meat up to 5 per cent by 2050.

Fig 5 shows the overall impacts of four types of protein systems under different scenarios. Scenarios 3, 2 and 1 have an overall positive impact value of 62, 60 and 46 (out of a possible score between −125 to +125), respectively. Based on the current impact assessment of 13 systems across 25 indicators, the overall score is + 7. It will improve with changes in the market share of various protein production systems till 2050, as shown in scenarios 1–3.

## Discussion

Our study makes three key contributions. First, ranking the 13 protein production systems based on a comprehensive set of multi-dimensional indicators reveals that regenerative beef production is a leading sustainable and responsible

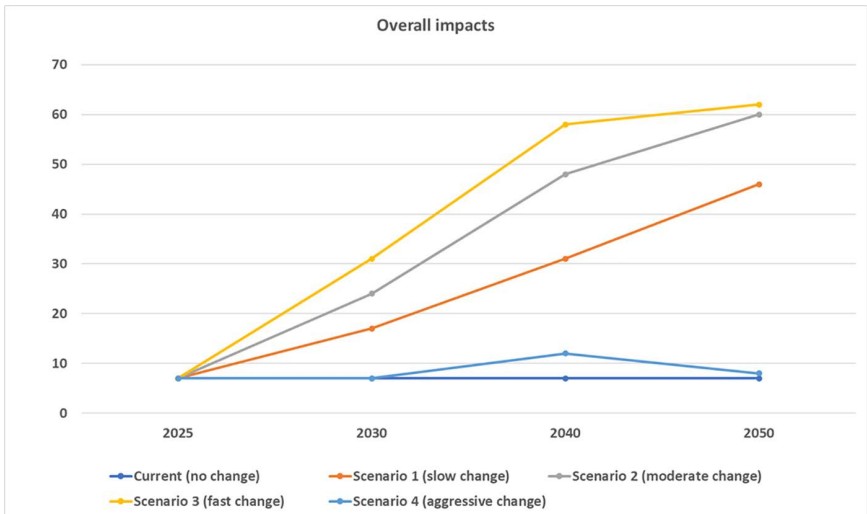

**Fig 5. Overall average impact of protein production systems from 2025 to 2050 based on five scenarios. Value: -125 to +125.**

system. Second, based on this impact assessment, we show that a just transition of the protein sector is possible with an increased market share of regenerative production systems (including organic, mixed livestock, rangeland systems, and small-scale production), plant-based protein, and cultivated meat by 2050. While the first two contributions are about impact assessment and its implications for the way forward towards a just transition of protein production systems, our third contribution is about advancing methodology. Through integrating a systematic review and the Delphi approach, our study advances the methodology required to generate reliable and valid expert consensus on diverse information and data. This methodology is particularly helpful in addressing complex problems that demand collating large amounts of data from diverse sources.

## Implications of impact assessment

Our analysis demonstrates that regenerative production systems with a holistic approach to raising animals based on rotational grazing and improved soil health showed positive impacts across all significant categories in this study. This is consistent with the current literature demonstrating that such systems have enormous potential to contribute to climate-resilient, equitable and economically sustainable protein production systems [45,46]. However, our study highlights that regenerative production systems need to reduce their impact on net GHG emissions, freshwater extraction and impacts on livestock diversity. Mixed, rangelands, free-range poultry, pastoralists, and plant-based systems also showed overall positive impacts. However, we found significant negative impacts on the natural capital of such systems. This is consistent with the growing literature on the large ecological footprint of animal agriculture in the last two decades and needs further attention from practice and policy [47–51]. Some of the natural capital indicators under plant-based systems, including indicator 2 (freshwater extraction—water pollution), indicator 6 (loss of agri-biodiversity on farm) and indicator 9 (soil contamination), have a lower score (−3) due to the association between plant-based food and on-farm primary production. This is because large amounts of lentils, grains, oil seed crops, etc., that form the key ingredients of highly processed plant-based food are grown using conventional farming methods. Therefore, as identified in this study, net GHG emissions are high and have high land use impacts.

Cultivated meat and organic systems positively impact several natural capital indicators, but the concerns about net GHG emissions, air pollution, energy use, and freshwater use remain unaddressed. Cultivated meat with conventional

and sustainable energy is ranked fifth and sixth, respectively. If the production of cultivated meat grows significantly in the future, it may effectively reduce the overall negative impacts of the protein production systems, alongside regenerative, organic, rangelands, and free-range poultry systems. Four systems—confined feeding operations, industrial pork, caged poultry, and small-scale beef—negatively impact natural capital the most [52]. Such systems focus only on high productivity and profits, negatively impacting animal welfare, farm workers' health and safety, and consumers' health, and pose grave concerns for achieving sustainable and equitable protein production systems [1,36].

## Implications for just transition

The scenarios developed in the study illustrate that continuous improvement of each system can move the sector towards sustainability and just transition. Unlike the energy sector, where just transition is defined as replacing the coal sector with renewable energy, one food system cannot simply be replaced with another preferred system [11,12]. This is because food systems are deeply embedded in culturally sensitive social structures and practices. Our study shows the pathway for just transition of the protein sector through various scenarios. Scenarios 1–3 show that if the currently dominant industrial production shifts to regenerative systems (including mixed, organic, rangelands, and small-scale) with a growing share of plant-based proteins and cultivated meat in the future, it can reduce the overall negative impacts on people and the planet. Just protein production systems must promote human development (e.g., through providing meaningful and profitable employment for livestock producers), enhance farm succession (e.g., by engaging younger generations in advanced agricultural practices and technologies), tackle climate change (e.g., through introducing regenerative and climate-conscious livestock production methods), and create avenues for the growth of novel innovations (e.g., plant-based alternatives and cultivated meat solutions) to address the challenges linked to the current lack of supply chain resiliency and food insecurity [53].

## Implications for policy and practice

The livestock and meat sector can use this comprehensive assessment of impacts to identify risks and develop appropriate responses in the value chain. For example, a large number of negative environmental, social, and health impacts of some protein production systems may lead to adverse regulatory responses that may attract heavy penalties, face consumer backlash, and become less attractive for future investments [54,55].

Similarly, livestock farmers and protein producers can target short-, medium- and long-term responses to avoid risks arising from negative impacts. Improved farming practices can address impacts identified on natural capital (by using renewable energy and changes in animal feed that reduce GHG emissions from animals), land use change (by avoiding forest clearing to raise animals), land degradation (by maintaining soil health), loss of biodiversity (by promoting local and traditional breeds of animals), air pollution (controlling odour and emissions of pollutants), and water use (recycling) [56]. Social capital impacts can be reduced at the community level with farmer cooperatives supported by appropriate agricultural and food policies [1,2]. Human capital impacts can be reduced by intervening at the government level to safeguard the health and safety of farmers and farm workers, providing appropriate training to the workforce, and promoting consumers' health [35]. Produced capital can be improved by increasing production efficiencies, fair prices and reduced input costs [55]. Improved governance of farms can also address negative impacts by implementing suitable policies to manage on-farm natural resources. Collective action by groups of livestock producers can also bring positive change in regional and national farm policies [57].

## Limitations

The limitations of our research offer novel directions for future exploration. We included the most relevant studies to develop an impact score within each of the five broad categories. However, a uniform distribution of literature in each category is not available. For example, for natural capital, many studies focus on GHG emissions and land use impacts

[47]. Livestock diversity and soil contamination are less represented. Similarly, consumers' health is well represented in the human capital literature because protein consumption is directly linked to health; however, there are fewer studies on workers' health and safety. In social capital, more studies focus on employment and livelihoods with less emphasis on community engagement and opportunities for young people. Future research should focus on these aspects of the protein production systems which are under investigated in the literature but are vital for its longer-term sustainability and just transition [45].

In addition, the majority of studies focus on beef production systems, followed by pork and poultry. As the production and consumption of plant-based alternatives grow, researchers must study their overall impacts on the environment and society [58]. There is no full-scale commercial production of cultivated meat to date; therefore, few studies relevant to cultivated meat were found in the literature. Recently, the United States Food and Drug Administration (FDA) and the United States Department of Agriculture (USDA) approved two cultivated meat companies to produce and market in the USA [59]. This will pave the way for the commercial production of cultivated meat. We encourage researchers to examine the social, economic and environmental sustainability in the future. This will add to the scientific literature, improving impact assessments and comparability of cultivated meat with other protein production systems.

The impacts based on a 10-point scale and subsequent scenarios should be used cautiously due to the methods' limitations. Interpreting a large amount of quantitative data and information on a comparable scale is subject to experts' understanding. We have minimised such concerns by engaging global experts to modify these impacts using the Delphi approach. Such methodology has also been used to arrive at a consensus in the medical and pharmaceutical industry, where multiple studies are conducted to test different drugs or procedures that can be used to treat the same disease [37]. Some indicators within the broad categories have the same score. This may lead to issues while drawing wider policy conclusions, as equal scores may overlook critical factors or overemphasise relatively less important indicators. Therefore, it is important to carefully consider the relative importance of each indicator when using this study for policy development. Practitioners, policymakers and industry should not cherry-pick selective indicators to advance a specific agenda and prefer one indicator over others. Instead, concerted efforts will be required to transform protein production systems towards sustainability by improving all relevant environmental, social and governance indicators.

The scenarios are limited in their scope as they explore only 'what if' scenarios based on the current and speculative market share of limited types of protein production systems. Many factors, such as financial models, investment, policy, technology, consumer preferences, inter-generational differences, climate change, and other known and currently unknown variables, may affect the outcome of this scenario analysis. The scenarios developed here are not prescriptive and should be used cautiously to explore a pathway towards just transition.

We focused on livestock-based protein systems and two new innovations (highly processed plant-based protein and cultivated meat). However, our study is also limited in that it does not include dairy, fish, and insects. There is extensive research on the role of dairy and fish in supplying protein, and there is growing literature on insect proteins [60–62]. Future research can explicitly focus on these other important sources of proteins.

We sought to represent the majority of global protein production systems but it is impossible to do field trials and bio-physical measurements to measure each indicator through controlled experiments to compare such diverse systems. Therefore, we relied on a systematic review of the literature and carried out a comprehensive and holistic analysis of impacts to establish a benchmark. As the meat and livestock industries, including plant-based alternatives and cultivated meat, grow to meet the global protein demand, they can respond to various impacts and work collaboratively to improve sustainability across five dimensions: environmental, economic, human, social, and governance. We would like to emphasise that our impact assessment intends to propose a pathway for just transition and not to single out any production system.

All protein production systems examined here have a negative impact on various indicators of natural capital. This is also highlighted in the literature, especially GHG emissions, biodiversity loss, and pollution [47,52]. Not surprisingly, we

found a higher number of studies focused on the impacts of protein production systems on natural capital in the literature analysis. This was followed by produced capital, because productivity and profits are important indicators affecting livestock farmers' incomes. Further research should focus on the social and human capital aspects of different production systems, as there are only a few studies in the current literature.

## Conclusions

This study provides a comprehensive overview of multi-dimensional impacts related to different protein production systems. The impacts of 13 protein production systems across 25 indicators under five key categories (natural capital, human capital, social capital, produced capital and governance) were analysed based on 285 studies. Nine protein systems (regenerative, organic, rangelands, free-range poultry, sustainable energy cultivated meat, conventional energy cultivated meat, mixed grains and livestock, pastoralists and plant-based) have overall positive impacts across all five categories. In comparison, four protein systems have negative impacts (small-scale beef, caged poultry, industrial pork, and confined feeding operations). This study shows that the regenerative production system is the best for the transition of protein production towards sustainability. However, negative impacts associated with greenhouse gas emissions, freshwater extraction and livestock diversity (natural capital) need to be addressed. Based on this analysis, future research could focus on improving various negative impacts of relevant indicators. Policy responses could support practices that can improve these indicators, reduce risks associated with negative impacts and assist in governing and managing protein production systems towards long-term sustainability. As demonstrated in the scenarios, rapid reduction of industrial production and improving each indicator by adopting sustainable practices in the meat sector can help achieve a just transition by 2050. A shift to regenerative production systems (including mixed, organic rangelands and small-scale) with an increasing share of plant-based and cultivated meat (sustainable energy) can help reduce the overall negative impacts on natural capital. Our impact assessment and the projected scenarios show a pathway to a diverse, resilient, and just transition of protein production systems.

## Supporting information

**S1 Table. Protein production systems and their descriptions.**
(DOCX)

**S2 Table. Impact categories and their definitions.**
(DOCX)

**S3 Data. Search strings and literature used in the analysis.**
(XLSX)

**S4 Table. A 10-point impact scale used in this study.**
(DOCX)

## Acknowledgments

The authors would like to thank the experts who contributed to the Delphi process.

## Author contributions

**Conceptualization:** Harpinder Sandhu, Amie Sexton, Lee Recht.

**Data curation:** Amie Sexton.

**Formal analysis:** Harpinder Sandhu, Priyambada Joshi.

**Funding acquisition:** Harpinder Sandhu, Lee Recht.

**Investigation:** Amie Sexton, Priyambada Joshi.

**Methodology:** Amie Sexton, Priyambada Joshi.

**Project administration:** Harpinder Sandhu, Amie Sexton.

**Resources:** Harpinder Sandhu, Lee Recht.

**Supervision:** Harpinder Sandhu, Ayon Chakraborty.

**Writing – original draft:** Harpinder Sandhu, Amie Sexton, Priyambada Joshi, Ayon Chakraborty, Sukhbir Sandhu.

**Writing – review & editing:** Harpinder Sandhu, Amie Sexton, Priyambada Joshi, Lee Recht, Ayon Chakraborty, Sukhbir Sandhu.

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
