## [Decision Letter · Decision Letter 0]

PONE-D-25-10342Impact assessment for just transition of protein production systemsPLOS ONE

Dear Dr. Sandhu,

Thank you for submitting your manuscript to PLOS ONE. After careful consideration, we feel that it has merit but does not fully meet PLOS ONE’s publication criteria as it currently stands. Therefore, we invite you to submit a revised version of the manuscript that addresses the points raised during the review process.

Please submit your revised manuscript by Jun 21 2025 11:59PM than this to complete your revisions, please reply to this message or contact the journal office at plosone@plos.org . Please include the following items when submitting your revised manuscript:

We look forward to receiving your revised manuscript.

Kind regards,

Andrey Nagdalian

Academic Editor

PLOS ONE

 [HS was awared the grant by Federation University Australia and Aleph Farms Joint Research grant G2355.

https://www.federation.edu.au/

https://aleph-farms.com/

Priyambada Joshi, PhD candidate received stipend/financial support from this joint research grant by Federation University Australia and Aleph Farms.]. 

Additional Editor Comments (if provided):

Reviewers' comments:

Reviewer's Responses to Questions

**Comments to the Author**

1. Is the manuscript technically sound, and do the data support the conclusions?

Reviewer #1: Partly

Reviewer #2: Yes

2. Has the statistical analysis been performed appropriately and rigorously? 

Reviewer #1: Yes

Reviewer #2: Yes

3. Have the authors made all data underlying the findings in their manuscript fully available?

Reviewer #1: Yes

Reviewer #2: Yes

4. Is the manuscript presented in an intelligible fashion and written in standard English?

Reviewer #1: Yes

Reviewer #2: Yes

5. Review Comments to the Author

Reviewer #1: The manuscript is technically sound, has sufficient sample sizes, controls and replicates that have undergone statistical processing. The work is presented using a large amount of data indicating the relevance of the development. Due to the fact that the relevance of this problem remains for a long time, for the appropriate conclusions, it is advisable to refer to modern works touching on similar topics, for example, in the following publications: DOI 10.1163/23524588-20230025; Development of a technology for the isolation and purification of insectoprotein from Zophobas morio biomass / I. V. Rzhepakovsky, P. A. Trushov, S. N. Povetkin, A. A. Naghdalyan // Modern science and innovation. - 2018. - No. 2 (22). - P. 85-91. - EDN VMYUHK.; Zophobas Morio Semi Industrial Cultivation Peculiarities / A. A. Nagdalian, S. V. Pushkin, I. V. Rzhepakovsky [et al.] // Entomology and Applied Science Letters. – 2018. – Vol. 5, No. 2. – P. 108-113. – EDN WFWHYJ.; Why does the protein turn black while extracting it from insects biomass? / A. A. Nagdalian, N. P. Oboturova, S. N. Povetkin [et al.] // Journal of Hygienic Engineering and Design. – 2019. – Vol. 29. – P. 145-150. – EDN POKELO.; https://doi.org/10.1051/e3sconf/202453703007. The text should be carefully checked for typos and grammatical errors.

Reviewer #2: Date: 24/04/2025. 

COMMENTS FOR THE EDITOR & AUTHORS. 

Dear Editors,

I have reviewed the manuscript "Impact assessment for just transition of protein production systems". The following minor points need to be addressed for further improvement of the article.

1.      The manuscript must include the recent research works on  application to protein production systems, especially from the past 5 years.2.      Introduction Section-Authors should justify here why this study is important and should state the uniqueness of this study.

3.      Please include the recent references for broad readers to understand the

importance of protein production from various sources and in the industrial applications.

4.      There are numerous grammatical errors and they significantly affect the flow of ideas. Please check through the MS.

5.      Please add line numbers for easy identification of specific lines in the manuscript during review.

6.   Research would be nice if you could include the latest updates on recent studies.

7. check thoroughly all typographical errors and references action should follow the journal guidelines.

8. conclusions need to elaborate and make more clarity.

6. PLOS authors have the option to publish the peer review history of their article (what does this mean? ). If published, this will include your full peer review and any attached files.

**Do you want your identity to be public for this peer review?** For information about this choice, including consent withdrawal, please see our Privacy Policy .

Reviewer #1: No

Reviewer #2: No

---

## [Author Response · Author response to Decision Letter 1]

29 May 2025

PONE-D-25-10342

Editor’s comments

Thank you for giving us the opportunity to revise our manuscript. We have addressed both the editor’s and reviewers’ comments below. Please refer to the Word file, Revised manuscript with track changes, for the changes mentioned below.

Comment Response

1 Format the manuscript using the PLOS ONE style Thanks for this comment. We have now carefully formatted the manuscript according to PLOS ONE's style requirements.

2 Financial disclosure We have provided this statement in the cover letter.

The authors would like to acknowledge the financial support provided by the Federation University Australia and Aleph Farms joint grant G2355. Priyambada Joshi (PhD candidate) received stipend/financial support from this joint research project funded by Federation University Australia and Aleph Farms. Lee Recht reports a relationship with Aleph Farms that includes employment. Co-author Dr Lee Recht was previously employed at Aleph Farms during the data analysis and writing the first draft of the article. Dr Recht has no longer worked at this organisation since August 2024. Other authors declare that they have no known competing financial interests or personal relationships that could have appeared to influence the work reported in this paper.

3 Include full ethics statement in the Methods section. Thanks for this comment. We have included it in the Methods section page 10, Delphi approach section. Lines 275-280.

4 Review reference list We have carefully checked all references cited in the paper and in the list. We have also revised the style of all references provided in the Supplementary Information and the Excel Database.

Response to reviewers’ comments

Reviewer 1

Comment Response

1 The manuscript is technically sound, has sufficient sample sizes, controls and replicates that have undergone statistical processing. The work is presented using a large amount of data indicating the relevance of the development. Thank you for your comment and encouragement.

2 Due to the fact that the relevance of this problem remains for a long time, for the appropriate conclusions, it is advisable to refer to modern works touching on similar topics, for example, in the following publications: DOI 10.1163/23524588-20230025; Development of a technology for the isolation and purification of insectoprotein from Zophobas morio biomass / I. V. Rzhepakovsky, P. A. Trushov, S. N. Povetkin, A. A. Naghdalyan // Modern science and innovation. - 2018. - No. 2 (22). - P. 85-91. - EDN VMYUHK.; Zophobas Morio Semi Industrial Cultivation Peculiarities / A. A. Nagdalian, S. V. Pushkin, I. V. Rzhepakovsky [et al.] //

Entomology and Applied Science Letters. – 2018. – Vol. 5, No. 2. – P. 108-113. – EDN WFWHYJ.; Why does the protein turn black while extracting it from insects biomass? / A. A. Nagdalian, N. P. Oboturova, S. N. Povetkin [et al.] //

Journal of Hygienic Engineering and Design. – 2019. – Vol. 29. – P. 145-150. – EDN POKELO.; https://doi.org/10.1051/e3sconf/202453703007.

Thank you for these helpful suggestions to include work on insect proteins. We have included them in the discussion section, limitations on page 20, lines 578-579. References 60-62.

Rzhepakovskiy IV, Trushov PA, Povetkin SN, Nagdalian AA. 2018. The development of technology of protein extraction and purification in Zophobas Morio. Modern Science and Innovations 2, 85-91. (In Russ.)

Nagdalian AA, Pushkin SV, Rzhepakovsky IV, Povetkin SN, Simonov A N, Verevkina MN, Ziruk IV. 2019. Zophobas Morio semi industrial cultivation peculiarities. Entomology and Applied Science Letters 5, 108-113

Nagdalian AA, Oboturova NP, Krivenko DV, Povetkin SN, Blinov AV, Verevkina MN, Marinicheva MP, Simonov AN, Rodin IA, Lakovets MG. 2019. Why does the protein turn black while extracting it from insect’s biomass? Journal of Hygienic Engineering and Design 29, 145-150.

3 The text should be carefully checked for typos and grammatical errors.

Thank you for this comment. The entire manuscript and the supplementary information have been proofread by a professional copy editor.

Reviewer 2

Comment Response

1 The manuscript must include the recent research works on application to protein production systems, especially from the past 5 years We thank the reviewer for this comment. As stated in the manuscript's methods section, we conducted a thorough systematic review of studies published until January 2025, including studies from 1986 onwards. In addition to these studies, we have included some recently published work on protein production in the introduction and conclusion to further strengthen our arguments.

See below for details on the various recent references being added to support our arguments.

Introduction: References 9,10,13,18

Discussion: References 46,51,60-62

2 Introduction Section-Authors should justify here why this study is important and should state the uniqueness of this study.

Thank you for this helpful comment. We now clearly describe the importance of the study and state its uniqueness in the introduction section (see page 3, lines 50-57). We have also included an updated reference to support this (13).

3 Please include the recent references for broad readers to understand the

importance of protein production from various sources and in the industrial applications. Thank you for your suggestion. The paper focuses on livestock-based protein production systems, with two protein innovations (plant-based and cultivated meat). However, as per the suggestions, we have included similar references from various sources, including industrial applications.

See new references, 9,10, 13,18,46,51, 60-62.

4 There are numerous grammatical errors and they significantly affect the flow of ideas. Please check through the MS.

Thank you for this comment. A professional copy editor has proofread the entire manuscript and the supplementary information.

5 Please add line numbers for easy identification of specific lines in the manuscript during review. We have added line numbers to the manuscript.

6 Research would be nice if you could include the latest updates on recent studies. Thank you for your suggestion. We have included the following references in the introduction and discussion sections to support our statements.

See new references, 9,10, 13,18,46,51, 60-62.

7 Check thoroughly all typographical errors and references, and the action should follow the journal guidelines. Thank you for this comment. A professional copy editor has proofread the entire manuscript and the supplementary information.

8 Conclusions need to be elaborated and made clearer.

Thank you for this suggestion. We have now added two main conclusions from our findings in the conclusion section.

See page 21, lines 604–614.

---

## [Decision Letter · Decision Letter 1]

Impact assessment for just transition of protein production systems

PONE-D-25-10342R1

Dear Dr. Harpinder Sandhu,

We’re pleased to inform you that your manuscript has been judged scientifically suitable for publication and will be formally accepted for publication once it meets all outstanding technical requirements.

Kind regards,

Andrey Nagdalian

Academic Editor

PLOS ONE

Reviewers' comments:

Reviewer's Responses to Questions

**Comments to the Author**

1. If the authors have adequately addressed your comments raised in a previous round of review and you feel that this manuscript is now acceptable for publication, you may indicate that here to bypass the “Comments to the Author” section, enter your conflict of interest statement in the “Confidential to Editor” section, and submit your "Accept" recommendation.

Reviewer #1: All comments have been addressed

Reviewer #2: All comments have been addressed

2. Is the manuscript technically sound, and do the data support the conclusions?

Reviewer #1: Yes

Reviewer #2: Yes

3. Has the statistical analysis been performed appropriately and rigorously? 

Reviewer #1: Yes

Reviewer #2: Yes

4. Have the authors made all data underlying the findings in their manuscript fully available?

Reviewer #1: Yes

Reviewer #2: Yes

5. Is the manuscript presented in an intelligible fashion and written in standard English?

Reviewer #1: Yes

Reviewer #2: Yes

6. Review Comments to the Author

Reviewer #1: The text should be carefully checked for typos and grammatical errors. The work is formatted using data indicating the relevance of the problem. The manuscript is technically sound, has sufficient sample sizes, controls and replicas that have undergone statistical processing.

Reviewer #2: I consider that the authors significantly improved their manuscript and I recommend to be published in the present form.

7. PLOS authors have the option to publish the peer review history of their article (what does this mean? ). If published, this will include your full peer review and any attached files.

**Do you want your identity to be public for this peer review?** For information about this choice, including consent withdrawal, please see our Privacy Policy .

Reviewer #1: No

Reviewer #2: **Yes: ** Govindarajan Rasiravathanahalli Kaveriyappan

---

## [Editor Report · Acceptance letter]

PONE-D-25-10342R1

PLOS ONE

Dear Dr. Sandhu,

I'm pleased to inform you that your manuscript has been deemed suitable for publication in PLOS ONE. Congratulations! Your manuscript is now being handed over to our production team.

Kind regards,

on behalf of

Dr. Andrey Nagdalian

Academic Editor

PLOS ONE